# Curcumin Induces Apoptosis of Chemoresistant Lung Cancer Cells via ROS-Regulated p38 MAPK Phosphorylation

**DOI:** 10.3390/ijms23158248

**Published:** 2022-07-26

**Authors:** Ming-Fang Wu, Yen-Hsiang Huang, Ling-Yen Chiu, Shur-Hueih Cherng, Gwo-Tarng Sheu, Tsung-Ying Yang

**Affiliations:** 1School of Medicine, Chung Shan Medical University, Taichung 40201, Taiwan; mfwu0111@gmail.com; 2Divisions of Medical Oncology and Chest Medicine, Chung Shan Medical University Hospital, Taichung 40201, Taiwan; 3Division of Chest Medicine, Department of Internal Medicine, Taichung Veterans General Hospital, Taichung 40201, Taiwan; waynehuang0622@gmail.com (Y.-H.H.); iang0810@gmail.com (L.-Y.C.); 4Institute of Biomedical Sciences, National Chung Hsing University, Taichung 40227, Taiwan; 5Faculty of Medicine, School of Medicine, National Yang Ming Chiao Tung University, Taipei 11221, Taiwan; 6Department of Biotechnology, Hung Kuang University, Taichung 43302, Taiwan; shcherng@sunrise.hk.edu.tw; 7Institute of Medicine, Chung Shan Medical University, Taichung 40201, Taiwan; 8Department of Life Sciences, National Chung Hsing University, Taichung 40227, Taiwan

**Keywords:** chemoresistance, curcumin, lung cancer, ROS, p38 MAPK

## Abstract

This study aimed to challenge chemoresistance by curcumin (CUR) with drug-selected human lung cancer A549 sublines that continuously proliferate in the present of docetaxel (DOC) and vincristine (VCR). Their sensitivities to CUR were measured by MTT assay and the particular intracellular reactive oxygen species (ROS) was detected by fluorescence activated cell sorting (FACS) analysis. Apoptosis was analyzed by Annexin V assay of the flow cytometry. Inhibitors and RNA interference were used to examine the signaling pathway regulated by the kinases. The obtained data demonstrated that CUR induces chemoresistant cell apoptosis by generating ROS and application of *N*-acetylcysteine (NAC) blocks ROS production, resulting in apoptosis suppression. Phosphorylation of extracellular regulated kinase (ERK), p38 MAPK, and eIF-2α were increased but c-Jun *N*-terminal kinase (JNK) did not increase when chemoresistant cells were treated with CUR. Downregulation of ERK and p38 MAPK phosphorylation by their inhibitors had no effect on CUR-induced apoptosis. Interestingly, the knockdown of p38 MAPK with shRNA significantly reduced CUR-induced apoptosis on the chemoresistant sublines. Phosphorylation of the eIF-2α protein was inhibited when p38 MAPK was knocked down in DOC-resistant A549 cells, but a high level of phosphorylated eIF-2α protein remained on the VCR-resistant A549 cells when p38 MAPK was knocked down. These data confirmed that CUR-augmented ROS potently induced apoptosis via upregulated p38 MAPK phosphorylation. Therefore, activated p38 MAPK is considered a pro-apoptotic signal for CUR-induced apoptosis of chemoresistant human lung cancer cells.

## 1. Introduction

Despite therapeutic advances in recent years, the effectiveness of standard anticancer treatment is limited by the development of drug resistance in cancer therapy. Chemotherapeutic drugs such as taxol/docetaxel or vinca alkaloids arrest cells and lead to mitotic catastrophe. Docetaxel (Taxotere^®^, DOC) has anti-mitotic properties by binding to microtubules (MTs) and stabilizing MTs [1], while vincristine (VCR) induces disruption of MTs by binding to tubulin and inhibiting tubulin polymerization [2]. Therefore, in contrast to DOC, the effect of VCR treatment is MTs destabilization. Both DOC and VCR drugs are a substrate of the ABCB1 transporter/p-glycoprotein (P-gp), and thus a high level of P-gp expression in cancer cells is predicted to develop drug resistance to DOC and VCR [3,4].

Natural compounds have been reported to modulate different survival pathways, and thus they can be used to prevent chronic diseases [5] and enhance the therapeutic effects of anticancer treatments [6]. Composites of selected anticancer drugs and small phytochemical molecules have recently emerged as a means of overcoming resistance in many cancers. Natural compounds such as curcumin (CUR) can modulate various survival routes to increase the curative effects of standard cancer treatments [7]. CUR is a phytopolyphenolic compound exhibiting anticancer effects in vitro and in vivo [8], as well as in human clinical trials [9]. In vitro studies reported the cytotoxicity of CUR on various cancer cells. Its cytotoxic potential is largely credited to the induction of apoptosis. CUR suppresses the growth of various cancer cells, including those from the pituitary gland, prostate, biliary, oral, and uterine leiomyoma cells [8]. The anticancer activity of CUR has been substantially demonstrated by orthotopic models or xenotransplantation in animal models, thus demonstrating its potential for treating lymphoma, melanoma, prostate, pancreatic, colorectal, hepatocellular, breast, ovarian, and bladder cancer [10].

Lung cancer is a major global health problem. Although long-established chemotherapy is still used as basic treatment for advanced lung cancer, it frequently fails with the development of drug resistance. When human non-small lung cancer (NSCLC) cell lines of A549 and H1299 were used, CUR induced apoptosis in both cell lines [11]. Later, it was reported that CUR caused DNA damage by promoting endoplasmic reticulum (ER) stress and apoptosis in A549 cells [12]. The production of reactive oxygen species (ROS) from CUR treatment led to caspase-dependent and caspase-independent apoptosis in mouse fibroblast L929 cells [13]. Further, CUR induced ROS-mediated anoikis, which was triggered by apoptosis also observed in other lung cancer H460 cells [14,15]. In sum, CUR has been demonstrated to mediate ROS generation, ER stress, and induce apoptosis in human lung cancer cells.

The members of the mitogen-activated protein kinases (MAPKs) family (extracellular regulated kinase (ERK), c-Jun *N*-terminal kinase (JNK), and p38 MAPK) have all been frequently associated with CUR activities through inactivation of these kinases. CUR decreased ERK1/2 activity in the PC-14 and H1299 human lung cancer cell lines associated with reduced survival and enhanced apoptosis [16]. Exposure of A549 and H1975 NSCLC cell lines to CUR could suppress mitomycin C-induced ERK1/2 signal activation and result in a synergistic cytotoxic effect [17]. CUR also inhibited the JNK pathway in LNCaP prostate cancer cells [18]. Further, CUR significantly suppressed the UVB-induced p38 MAPK and JNK phosphorylation of human foreskin obtained from dermal fibroblasts [19]. In contrast to the downregulation of MAPKs by CUR, upregulation of MAPKs by CUR has also been reported by other researchers. The apoptosis of the human retinoblastoma cell line Y79 can be induced by CUR through activating the JNK and p38 MAPK pathways. Further treatment of Y79 cells with JNK and p38 MAPK inhibitors significantly suppressed the CUR-induced activation of caspases [20]. CUR-induced apoptosis and oxidative stress were associated with phosphorylation and activation of JNK, p38, and ERK in A549 cells [21]. Therefore, the MAPKs signaling pathway is closely associated with CUR activities but the regulation of MAPKs by CUR has to be extensively determined in each issue.

This study examined whether CUR has a notably cytotoxic effect on chemotherapeutic drug selected lung cancer A549 cells that have different P-gp expression levels. We also investigated the MAPK signaling pathway to identify the critical mediator for CUR-induced apoptosis on the chemoresistant lung cancer cells. The essential role of ROS induced endoplasmic reticulum (ER) stress has also been examined.

## 2. Results

### 2.1. Curcumin Induces Significant Apoptosis of Chemoresistant Lung Cancer Cells without an Additional Toxic Effect when Combined with Chemotherapy 

To evaluate the cytotoxicity of CUR, we used lung cancer A549/D16 cells that had been selected in the long-term by DOC and could be cultured with 16 nM of DOC. Another cell named A549/V16 that had been selected by VCR could proliferate with 16 nM of VCR. Both chemoresistant sublines were established previously and had been demonstrated to have multi-resistance to DOC, VCR, and doxorubicin [22]. The high level of P-gp expression of A549/D16 cells provides a useful marker to examine whether P-gp reduces the effect of CUR on chemoresistant lung cancer cells. Further, the chemoresistance of A549/V16 cells was independent of their P-gp level. To examine the possible interaction between CUR and chemotherapy, DOC and CUR were combined to treat A549/D16 cells (Figure 1A), and VCR was combined with CUR to treat A549/V16 cells (Figure 1B). The parental A549 cells were used as a control to test the effectiveness of chemotherapy in each in vitro experiment. Cell viabilities analyzed by MTT assay were significantly reduced when the A549 cells responded in a dose dependent manner to CUR. Apparently, only the parental A549 cells responded to chemotherapy; approximately 56% and 66% of the A549 cells survived when DOC (16 nM) or VCR (16 nM) were applied, respectively. Adding various concentrations of CUR to DOC or VCR further reduced A549 cell survival. In contrast to their parental cells, the chemoresistant A549/D16 and A549/V16 cells were not affected by chemotherapy. When CUR was combined with either chemotherapy to treat A549/D16 or A549/V16 cells, the cell viabilities were not significantly changed when compared with CUR alone. There was no additional cytotoxicity found, nor was an antagonism or sensitization effect observed. These results indicated that CUR alone can inhibit survival of parental A549 cells and chemoresistant sublines effectively. Whether CUR is essential to induce apoptosis of chemoresistant lung cancer cells without chemotherapy was further examined. The A549/D16 cells were treated with CUR at 20, 30, and 40 μM concentrations for 48 h, followed by Annexin V apoptotic assay to examine the ratio of apoptosis induced by CUR (Figure 1C). Early apoptotic cells (lower right square, LR) increased from 0.59% (control) to 1.68%, 9.56%, and 24.49%, respectively, when the CUR concentration was increased. The late apoptotic cells (upper right square, UR) increased from 0.77% to 1.12%, 5.58%, and 19.14%, respectively (Left panel). The right panel also shows that cells treated with more than 30 μM of CUR resulted in significant apoptosis of chemoresistant A549/D16 cells. Similar treatments were also applied to A549/V16 cells to determine the effect of CUR and higher apoptotic ratios were obtained (Figure 1D). The early and late apoptotic A549/V16 cells increased from 3.5% to 9.98%, and 56.14% and 77.86%, respectively, with CUR dose dependence. The proteins of cleaved caspase 7, PARP and c-PARP were examined by western blot assay (Figure 1E,F) to verify the activation of caspase 7 and completion of apoptosis when the PARP protein was cleaved. Caspase 7 is one of the executioner caspases (caspase 3, 6, 7) cleaved and activated by initiator caspases, regulating the apoptotic pathway [23].

### 2.2. Pan-Caspase Inhibition Blocks Curcumin-Induced Apoptosis in Chemoresistant Lung Cancer Cells

The apoptosis of chemoresistant lung cancer cells was further analyzed by pan-caspase inhibitor (Z-VAD-FMK, 50 μM) pretreatment followed by Annexin V apoptotic assay. The results of CUR induced apoptosis on the A549/D16 cells were similar to the results shown in Figure 1C. Combined early and late apoptosis induced by 40 μM of CUR was 42.67%. However, after the pan-caspase activity was blocked, the combined apoptosis was significantly reduced to 2.95% in the A549/D16 cells (Figure 2A). The apoptosis was also significantly reduced from 78.38% to 7.97% when the pan-caspase inhibitor was applied to the A549/V16 cells (Figure 2B). The cleaved PARP levels were significantly reduced when both sublines were pretreated with Z-VAD-FMK, demonstrating that the caspases’ activities were repressed by Z-VAD-FMK (Figure 2C,D). 

### 2.3. Curcumin Induced Apoptosis by Augmenting Reactive Oxygen Species Production That Can Be Relieved with NAC 

It has been reported that CUR induced apoptotic cell death in human non-small cell lung cancer cells (NCI-H460) with significantly increased reactive oxygen species (ROS) [15]. To analyze the amount of ROS, we applied 10 μM of fluorescein H2DCFDA to detect the ROS generated upon different concentrations of CUR treatment applied to chemoresistant sublines (Figure 3A,B). Compared to cells without CUR treatment, increased ROS was detected with dose-dependence in both sublines and CUR treatment induced more ROS in A549/V16 cells than A549/D16 cells. The association between ROS and apoptosis was further investigated with the pretreatment of 10 mM NAC (ROS scavenger) to suppress the ROS level (Figure 3C,D). The amount of ROS generated by CUR (blue peak) was significantly reduced with NAC pretreatment (red peak) and NAC alone has a similar basic level of ROS (brown peak) when compared with the mock cells (black peak). The effect of NAC on CUR-induced apoptosis was examined by Annexin V apoptotic assay in A549/D16 cells (Figure 3E) and A549/V16 cells (Figure 3F). The data show that NAC blocks the apoptosis significantly in both sublines when cells are exposed to CUR.

### 2.4. Association of ERK, p38 MAPK and eIF2-α Phosphorylation with Curcumin Induced Apoptosis in Chemoresistant Lung Cancer Cells

In addition to lung cancer cells, CUR also induced ROS production in human gastric cancer BGC-823 cells by activating the JNK signaling pathway [24]. In Y79 human retinoblastoma cells, CUR induced the apoptosis of cells through the JNK and p38 MAPK pathways [20]. Interestingly, CUR also inhibited the JNK pathway in LNCaP prostate cancer cells [18] and human hepatic LO2 cells [25]. Therefore, to distinguish how the MAPK signaling pathways were regulated by CUR in lung chemoresistant cells, we further investigated the MAPKs signaling pathways by western blot analysis on A549/D16 cells (Figure 4A) and A549/V16 cells (Figure 4B). The levels of phosphorylated ERK and p38 MAPK, but not JNK, were significantly enhanced when the chemoresistant A549 sublines were treated with CUR. Phosphorylated forms of eIF2α (to attenuate translation initiation) were increased, which was correlated with the CUR concentration and c-PARP levels. When NAC was applied to the chemoresistant A549 sublines before CUR treatment, the MAPK signaling pathways were not activated and eIF2α phosphorylation was terminated, resulting in an anti-apoptotic effect.

### 2.5. Anti-Apoptotic Effect Was Not Obtained by Pretreating with ERK or p38 MAPK Inhibitors

To examine the effect of ERK activation on CUR-induced apoptosis, cells were pretreated with the ERK inhibitor (U0126). This was followed by adding CUR to the A549/D16 (Figure 5A) and A549/V16 cells (Figure 5B) for apoptosis analysis. The early and late apoptotic A549/D16 cells treated with CUR (40 μM) were not significantly reduced when U0126 was applied. The proteins of ERK, p-ERK, and c-PARP were examined by western blot assay (Figure 5C) to verify the effects of ERK inhibition and apoptosis. Although the p-ERK levels were reduced by U0126 treatment, the c-PARP levels were not significantly reduced in either subline. The data indicated that ERK activation was not associated with curcumin-induced apoptosis. Therefore, it would be very interesting to know if p38 MAPK plays a crucial role in CUR-induced apoptosis. Both chemoresistant sublines were pretreated with the p38 MAPK inhibitor (SB203580) and then exposed to CUR, with the results shown in Figure 6. Surprisingly, the levels of apoptosis were not reduced by SB203580 inhibition, the A549/D16 cells (Figure 6A) nor A549/V16 cells (Figure 6B). When we carefully compared the inhibition efficiency of SB203580 on p38 phosphorylation, the protein detection data (Figure 6C) further showed the dose-related CUR induced p-p38 levels were not observed when SB203580 was used. It should be noted that SB203580 treatment somehow enhanced the basic level of phosphorylated p38 MAPK. We thought that we may require a better method to block p38 phosphorylation in order to detect its phosphorylation effect on apoptosis. To have greater efficiency in inhibiting p38 MAPK activation, we applied RNA interference using shRNA to specifically knockdown p38 MAPK.

### 2.6. Knockdown of p38 MAPK Significantly Reduced CUR-Induced Apoptosis

When A549/D16 cells were treated with CUR (40 μM), knockdown of p38 MAPK by shp38-10052 reduced the early apoptosis from 18.92% to 6.01%, and late apoptosis from 6.63% and 1.36%, when compared with the transduction control of shLUC (Figure 7A). Similar results were also obtained from experiments using A549/V16 cells that reduced early and late apoptosis from 30.25% to 17.95%, and 57.09% to 7.77%, respectively (Figure 7B). The proteins of p38 MAPK, eIF2α, and PARP were examined by western blot assay to confirm the inhibition of p38 MAPK phosphorylation and reduced apoptosis (Figure 7C,D). The data showed the protein levels of p38 MAPK were merely detected in both sublines and the phosphorylation of eIF2α was blocked in the A549/D16 cells but not in the A549/V16 cells. When p38 MAPK knockdown was achieved, the cleaved PARP levels were reduced that coordinated with the CUR-induced apoptosis. Further, p38 MAPK was found to be an upstream mediator of eIF2α phosphorylation in A549/D16 cells. However, this activity was not found in A549/V16 cells.

## 3. Discussion

The antioxidant activity of CUR has been repeatedly demonstrated in numerous clinical trials [26,27]. CUR is an effective scavenger of ROS, which are generated in excess due to environmental influences [28]. In contrast to its antioxidant activity, higher than micro-molar concentrations of CUR inducing ROS were documented with significant anticancer activities in many in vitro models of cancer cells as well as in animal models and clinical trials [9,29]. Therefore, CUR can both decrease and increase the cellular levels of ROS that may rely on its concentration at the target site and the affected cells. Accordingly, CUR use as a chemopreventive or chemotherapeutic agent should be carefully defined, as has been previously suggested [30]. 

Chemotherapeutic agents of cisplatin and doxorubicin are ROS-inducing anticancer drugs [31]. Their activity also can be found in the vinca alkaloids of VCR [32] and taxanes of DOC [33,34]. The current study is the first to examine the efficacy of CUR on DOC/VCR multi-resistant lung cancer cells that acquire their resistance from chemotherapeutic drugs. 

Our data reveal that CUR alone has significant cytotoxicity in parental lung cancer A549 cells, which is in accordance with previous reports [35,36], and that CUR increases the DOC/VCR toxicity of A549 cells in a dose-dependent manner (Figure 1). Since CUR cannot alleviate the efficacy of chemotherapy, CUR is therefore not acting as an antioxidant against those ROS generated by chemotherapeutic drugs on A549 cells. Although CUR alone can reduce DOC- and VCR-resistant cell survival, the combination of chemotherapy with CUR has little effect in further decreasing the survival of chemoresistant cells. These data indicate that CUR at the indicated concentrations cannot reverse the chemoresistance to sensitize the A549/D16 and A549/V16 sublines to chemotherapy and that the cytotoxicity of CUR is not associated with the level of P-gp expression. Further, chemoresistant cells have similar CUR sensitivities when compared with their parental A549 cells. The results provide two potential therapeutic applications of CUR. The first involves combining CUR with chemotherapy in first-line chemotherapy to enhance the efficacy of lung cancer therapy and possibly reducing the deleterious effect of chemotherapy [37,38]. The second involves administering CUR in refractory lung cancer without chemotherapy, which has not been generally recognized. The critical problem for these applications is how to reach a maximal dosage of CUR at the target site when the bioavailability of CUR is low. Therefore, the question concerning how to meet this challenge has been heavily discussed [9,39] and promising future results are expected.

CUR inducing caspase-dependent apoptosis of chemoresistant lung cancer cells is demonstrated in Figure 2. The pro-oxidant role of CUR in chemoresistant lung cancer cells is clearly shown in Figure 3, whereby NAC potently prevents CUR-induced ROS and leads to reduced apoptosis. When the MAPKs signaling pathways were analyzed, the upregulated phosphorylation of the ERK and p38 MAPK proteins were observed (Figure 4). Interestingly, neither the ERK (U0126) nor p38 MAPK (SB203580) inhibitors affect CUR-induced apoptosis. U0126 effectively inhibits ERK phosphorylation, indicating ERK activity was significantly reduced by U0126. On the other hand, p38 MAPK maintained a high phosphorylation status when SB203580 was applied (Figure 6C). The data indicated p38 MAPK might be constitutionally activated by CUR and only partially inactivated by SB203580. Interestingly, temperately inactivated p38 MAPK by its inhibitor was insufficient to prevent CUR-induced apoptosis. To significantly inhibit p38 MAPK, a knockdown assay by p38 MAPK-specific shRNA was performed. The results in Figure 7 showed that p38 MAPK is a critical mediator of CUR-induced apoptosis. 

It is widely recognized that ERK generally reacts to growth stimulation, while JNK and p38 MAPK activate in response to exogenous factors, such as inflammation, UV irradiation, DNA damage, and oxidative stress [40]. The canonical activation pathway of p38 MAPK phosphorylation at the Thr180 and Tyr182 residues is essential to activate its kinase activity to either prevent or induce apoptosis [41]. The intensity of p38 MAPK activation has been linked with its pro-death or pro-survival role, whereby potent p38 MAPK activation is associated with senescence [42] and terminal cell differentiation [43]. On the other hand, normal p38 activation has a cell survival effect [44]. Our data provided novel evidence that CUR induces strong activation of p38 MAPK as a pro-death signal for apoptosis of chemoresistant cells. 

In addition to p38 MAPK being activated by CUR, phosphorylation of eIF2α is also upregulated. When ROS was prevented by NAC, the eIF2α was not phosphorylated (Figure 4), suggesting ROS is an upstream regulator of eIF2α in CUR-treated chemoresistant cells. Generally, the ER stress transducer protein kinase RNA-like ER kinase (PERK) is activated by ER stress to phosphorylate eIF2α and result in extensive repression of translation [45]. Phosphorylation of eIF2α on serine 51 is known to be mediated by four upstream eIF2 kinases: PERK, protein kinase RNA-activated (PKR), general control non-derepressible 2 (GCN2), and heme-regulated inhibitor (HRI) [46]. Sustained and unsolved ER stress motivates the programed cell death of apoptosis [47,48]. We have not yet characterized how p38 MAPK interacts with each of the abovementioned kinases to mediate CUR-induced phosphorylation of eIF2α in A549/D16 cells. In A549/V16 cells, knockdown of p38 MAPK could not repress eIF2α phosphorylation (Figure 5), suggesting that CUR induces another signaling pathway independent of p38 MAPK activation and potentiates ER stress in A549/V16 cells. 

The following question must be addressed: How do other pathways, in addition to the reported MAPK pathway, affect the chemoresistance that attenuated by CUR? We believe that CUR diminishes cancer stemness also play a critical role on chemoresistance. The potential of CUR to regulate the growth of cancer stem cell (CSC) has been reviewed by Sordillo and Helson [49]. Suppression of the cytokines, interleukin (IL)-6, IL-8, IL-1, and CSC pathways, such as Wnt, Notch, Hedgehog and FAK by CUR were indicated. It has also been reported that CUR inhibited lung cancer stem cell traits and prohibited tumorsphere formation [50]. In a recent report, it has been shown CUR (125 nM) treatment reduced the sphere formation ability at the concentrations would not affect the cell viability of A549 cells and normal pulmonary epithelial cells [51]. Therefore, the multi-potential of CUR with anticancer abilities warrants further experimental study to identify its optimal application.

Another question must be addressed: How does CUR interact with each specific signal pathway? Moustapha et al. [52] have analyzed that Huh-7 liver cancer cells treated with CUR (up to 80 μM) for 5 min and followed by intracellular CUR concentration measurement. Their data have shown that CUR enters cells rapidly to a final ratio between external added CUR and the intracellular concentration of 1/20. It corresponds to 1μM of CUR as the intracellular concentration for 20μM added externally. Later, the same group of researchers reported that CUR localization at the endoplasmic reticulum (but not on the mitochondrial network) caused an unfolded protein response (UPR) and affected calcium status [53]. Other methods of molecular interaction of CUR with various cellular proteins have been reviewed by Gupta et al. [54]. Interestingly, a computational study for the possibility of CUR to bind the coronavirus (SARS-CoV2) viral spike protein (S Protein) and the cognate host cell receptor angiotensin-converting enzyme 2 (ACE2) has been reported [55]. By using a molecular simulation study, CUR directly binds to the receptor binding domain (RBD) of viral S Protein and host ACE2, thus interfering with the formation of S Protein-ACE2 complex. The data provide a potential link for how CUR interacts with membrane receptor for further signaling.

## 4. Materials and Methods

### 4.1. Drugs and Chemicals

DOC was obtained from Aventis Pharmaceuticals Inc. (Bridgewater, NJ). VCR and *N*-acetylcysteine (NAC, A-7250) were purchased from Sigma-Aldrich (St. Louis, MO, USA). Z-VAD-FMK was purchased from Bachem (Torrance, CA, USA). U0126 and SB203580 were purchased from Cell Signaling (Danvers, MA, USA). Curcumin was obtained from Cayman Chemical (Ann Arbor, MI, USA). PhosSTOP and cOmplete Tablets EASYpack were purchased from Roche (Basel, Switzerland).

### 4.2. Chemoresistant Subline and Viability MTT Assay

Human adenocarcinoma A549 cells (ATCC, Manassas, VA, USA) were maintained as previously described [22]. DOC/A549 and VCR/A549 resistant cells were generated from parental cells by exposure to increasing concentrations of DOC or VCR in a stepwise manner. The DOC-resistant subline maintained at 16 nM DOC is denoted A549/D16 and is identified as P-gp overexpressing cells. A similar designation, A549/V16, was given to a VCR stably resistant subline and this subline was P-gp-independent [22]. Cell viabilities were determined on MTT colorimetric assay. Briefly, cells (2 × 10^4^/per well) were seeded onto 24-well plates. After 24 h incubation, the cells were exposed to various concentrations of DOC or VCR or CBCs in fresh medium for 48 h. Cells were washed with PBS, and MTT (300 μL/well, 1 mg/mL; Sigma) was added prior to further incubation at 37 °C for 2.5 h. Cells were washed with PBS, and 2-propanol solution (300 μL/well) was added to dissolve the water-insoluble formazan salt by shaking at 70 rpm for 10 min at room temperature. At last, the absorbance (570 nm) was detected by an ELISA plate reader (Molecular Devices SPECTRA max 340 PC).

### 4.3. Detection of Intracellular ROS by Fluorescence Activated Cell Sorting (FACS) Analysis

Cells of A549 (1 × 10^5^ cells) were seeded in 6-well plates for 16 h, followed by CUR treatment. Alternatively, pretreatment with or without 10 mM NAC for 1 h to inhibit ROS was then followed by CUR treatment. Later, PBS was used to wash the cells and the cells were harvested by trypsin treatment. Those cells were then resuspended with 0.5 mL PBS containing 10 μM of 2′, 7′-dichlorodihydro-fluorescein diacetate (D399, H2DCFDA, Invitrogen) fluorescent probe for 30 min. The cells were collected and immediately analyzed by a flow cytometer.

### 4.4. Western Blot Analysis

The apoptosis and signaling pathway regulated proteins were analyzed. The relevant procedures were previously described [22]. Proteins were reacted with one of the following antibodies for ERK, phosphos-ERK (p-ERK), p38 MAPK, phosphos-p38 MAPK (p-p38), JNK, phosphos-JNK (p-JNK), Poly (ADP-ribose) polymerase (PARP), eIF-2α, phosphos-eIF2α (ser51), cleaved caspase 7 purchased from Cell Signaling (Danvers, MA), and GAPDH purchased from Proteintech (Rosemont, IL, USA).

### 4.5. Annexin V Assay for Apoptosis Characterization

Cells (1 × 10^5^ cells) were seeded in 6-well plates for 16 h followed by CUR treatment. Alternatively, they were pretreated with or without 10 mM NAC for 1 h to inhibit ROS, followed by CUR treatment. The cells were trypsinized and incubated for 30 min in binding buffer with propidium iodide (PI) and Annexin V (FITC Annexin V Apoptosis Detection Kit 1, BD Biosciences, San Jose, CA), followed by analysis with flow cytometry.

### 4.6. RNA Interference of p38 MAPK

The RNA interference was performed by lentiviral delivery of gene-specific short hairpin RNA (shRNA). Lentiviral vectors carrying human p38 MAPK specific shRNA (shp38-10052, TRCN0000010052) and transduction control (shLUC, TRCN0000072249) were constructed and packed by the National RNAi Core Facility at the Institute of Molecular Biology (Academia Sinica, Taipei City, Taiwan). For virus transduction, cells were seeded at 5 × 10^5^ cells per well in 60-mm plates and transduced with lentivirus for 24 h. The transduced cells were treated with 2 g/mL puromycin for 48 h for further applications.

### 4.7. Statistical Analysis

All values are presented as mean ± SD. Data were compared between the indicated groups using t-test and * *p* < 0.05 was considered statistically significant.

## 5. Conclusions

The general summary provided in Figure 8 concludes that CUR, at a high concentration, is an effective pro-oxidant molecule to generate high levels of intracellular ROS to induce caspase-dependent apoptosis of chemoresistant lung cancer cells. The p38 MAPK signaling pathway activated by CUR also regulates eIF2α phosphorylation in A549/D16 cells but other signaling pathways mediate eIF2α phosphorylation in A549/V16 cells. Therefore, potent p38 MAPK activation is essential to induce apoptosis of chemoresistant lung cancer cells.

## Figures and Tables

**Figure 1 ijms-23-08248-f001:**
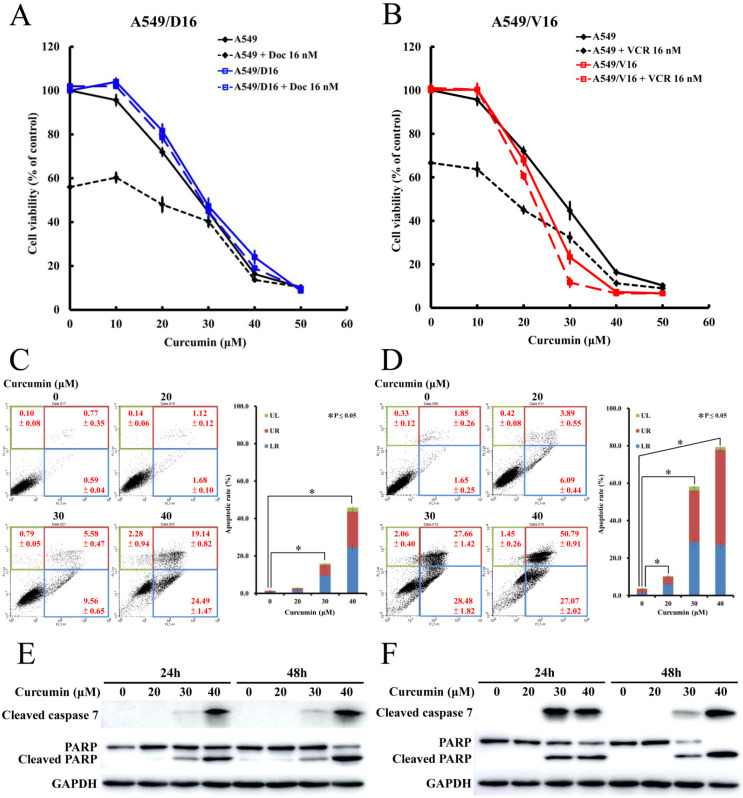
Curcumin induces apoptotic cell death in chemoresistant lung cancer A549 cells. The cells were exposed to curcumin alone or combined with chemotherapeutic drugs (Doc or VCR) for 48 h followed by MTT assay. (**A**) Black color lines indicate the data obtained from parental A549 cells, and blue color lines indicate the data obtained from chemoresistant A549/D16 cells. (**B**) Red color lines indicate the data obtained from A549/V16 cells. The levels of apoptosis were analyzed with flow cytometry by staining with propidium iodide (PI) and Annexin V. on (**C**) A549/D16 cells and (**D**) A549/V16 cells. The cleaved caspase 7 and PARP were detected by Western blot analysis on (**E**) A549/D16 cells and (**F**) A549/V16 cells to confirm the apoptosis was induced by curcumin. (lower right square, LR; upper right square, UR; upper left square, UL).

**Figure 2 ijms-23-08248-f002:**
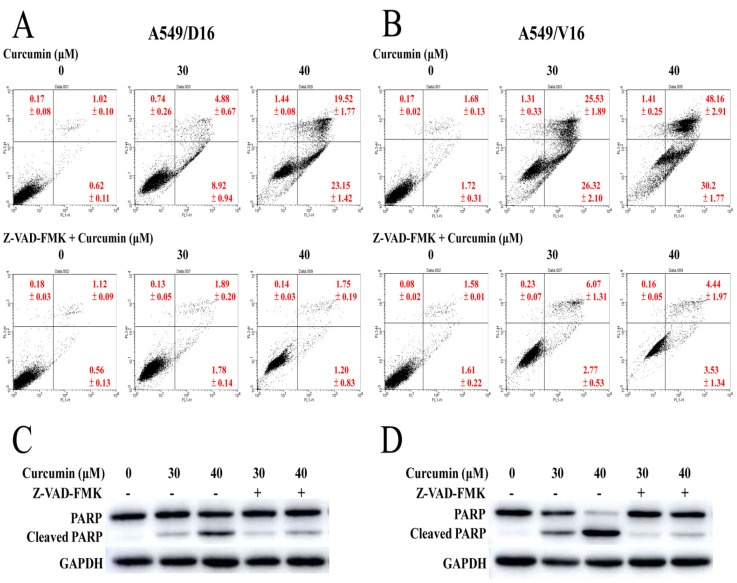
Application of pan-caspase inhibitor to chemoresistant cells for examining curcumin-induced apoptosis. Chemoresistant cells were pretreated with pan-caspase inhibitor Z-VAD-FMK (50 μM) for 1 h followed by exposure to various concentrations of CUR for 48 h. (**A**) Analysis of apoptosis with PI/Annexin V staining by flow cytometry on A549/D16 cells and (**B**) A549/V16 cells. The levels of cleaved PARP were observed by Western blot analysis in (**C**) A549/D16 cells and (**D**) A549/V16 cells.

**Figure 3 ijms-23-08248-f003:**
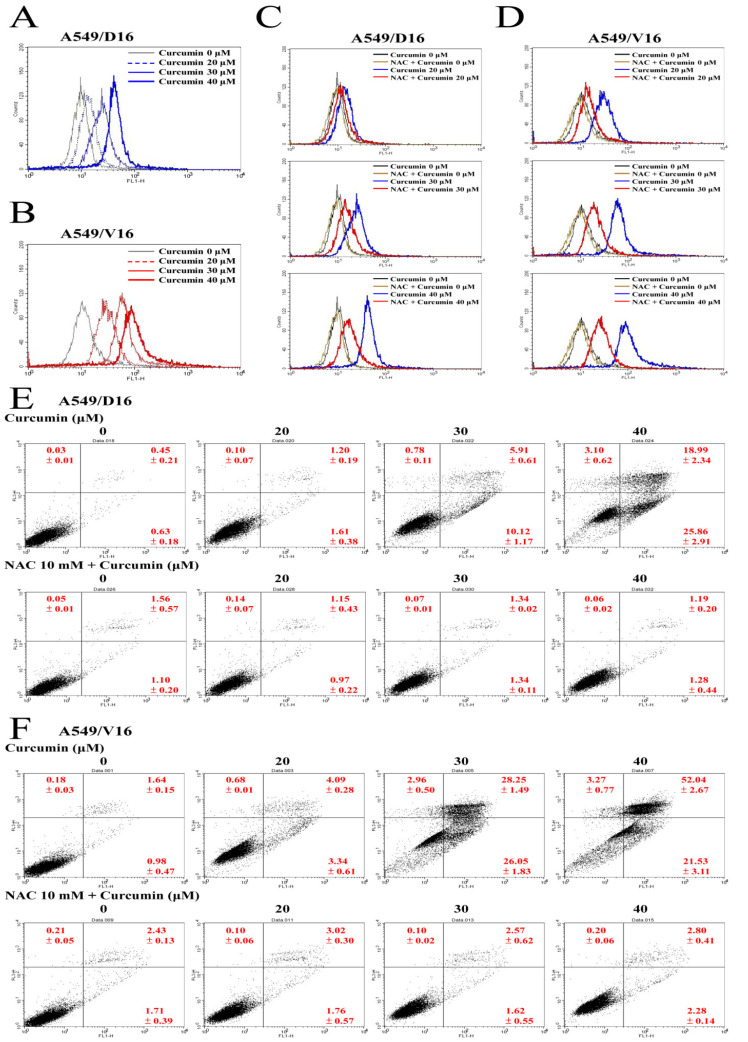
Examination of curcumin induced ROS-mediated apoptosis with FACS analysis. Cells were incubated with 10 μM of H2DCFDA fluorescent probe for 30 min. (**A**) 549/D16 and (**B**) A549/V16 cells were then washed with PBS, trypsinized and immediately analyzed by a flow cytometer to measure their ROS level. Similar experiments were performed but (**C**) A549/D16 and (**D**) A549/V16 cells were pretreated with or without 10 mM NAC (1 h), followed by curcumin treatment (48 h) for ROS detection. NAC inhibited apoptosis of (**E**) A549/D16 and (**F**) A549/V16 cells were analyzed with flow cytometry by staining with PI/Annexin V.

**Figure 4 ijms-23-08248-f004:**
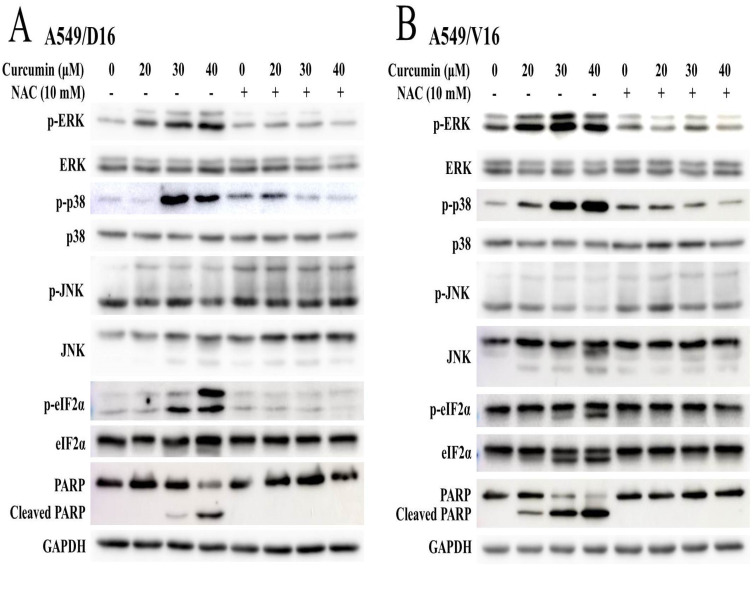
Association of the MAPK signaling pathways with curcumin induced ROS, ER stress and apoptosis. Chemoresistant cells of (**A**) A549/D16 and (**B**) A549/V16 were pretreated with or without antioxidant NAC followed by curcumin treatment. The protein of ERK, p38 MAPK and JNK and their phosphorylated forms were detected by western blots. The cleaved PARP was used to mark the apoptosis event and the level of phosphorylated e-IF2α was used to correlate ER stress. GAPDH was used as the control of total protein loaded.

**Figure 5 ijms-23-08248-f005:**
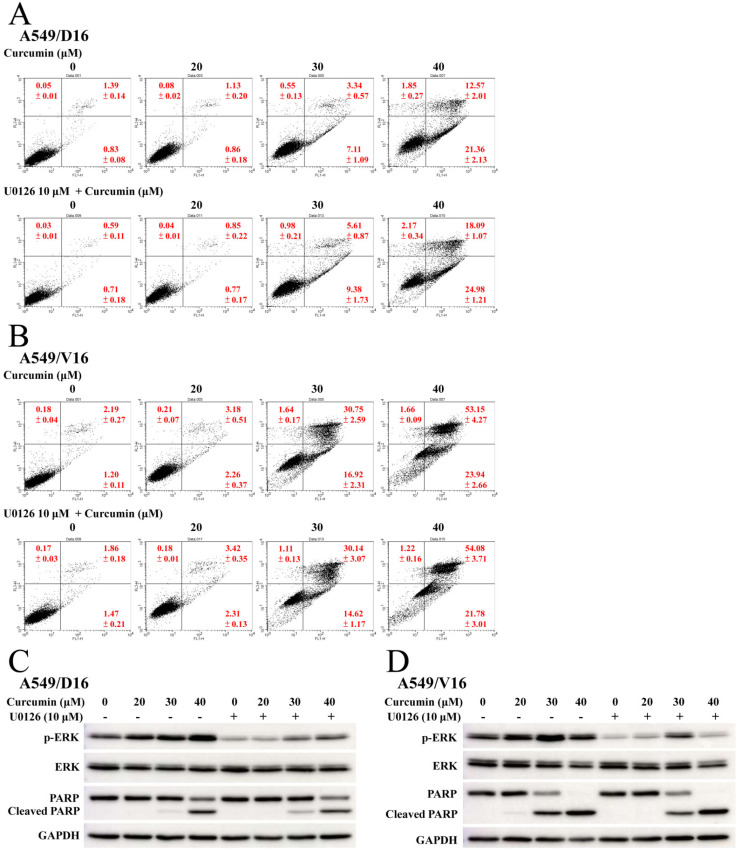
Inhibition of ERK by U0126 failed to reduce curcumin-induced apoptosis. Chemoresistant cells were pretreated with or without ERK inhibitor U0126 (10 μM) for 1 h followed by CUR treatment for 48 h. (**A**) Analysis of apoptosis with PI/Annexin V staining of flow cytometry for A549/D16 cells and (**B**) A549/V16 cells. (**C**) The levels of ERK, p-ERK, and cleaved RARP were detected by Western blot analysis of the A549/D16 and (**D**) A549/V16 cells.

**Figure 6 ijms-23-08248-f006:**
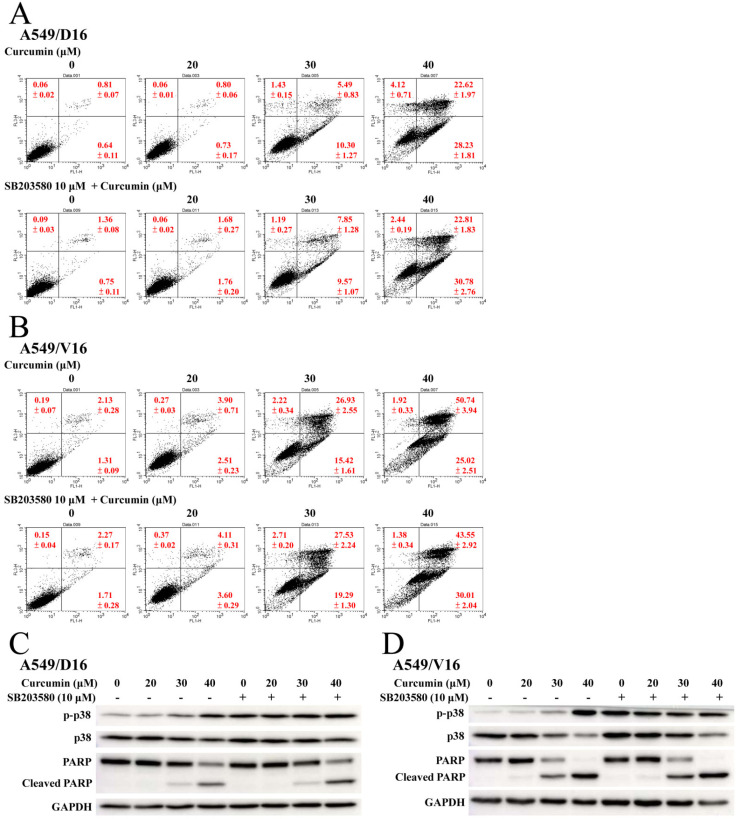
Inhibition of p38 kinase by SB203580 failed to reduce curcumin-induced apoptosis. Chemoresistant cells were pretreated with or without p38 kinase inhibitor SB203580 (10 μM) for 1 h followed by CUR treatment for 48 h. (**A**) Analysis of apoptosis with PI/Annexin V staining of flow cytometry for A549/D16 cells and (**B**) A549/V16 cells. (**C**) The levels of ERK, p-ERK and cleaved RARP were detected by western blot analysis of the A549/D16 and (**D**) A549/V16 cells.

**Figure 7 ijms-23-08248-f007:**
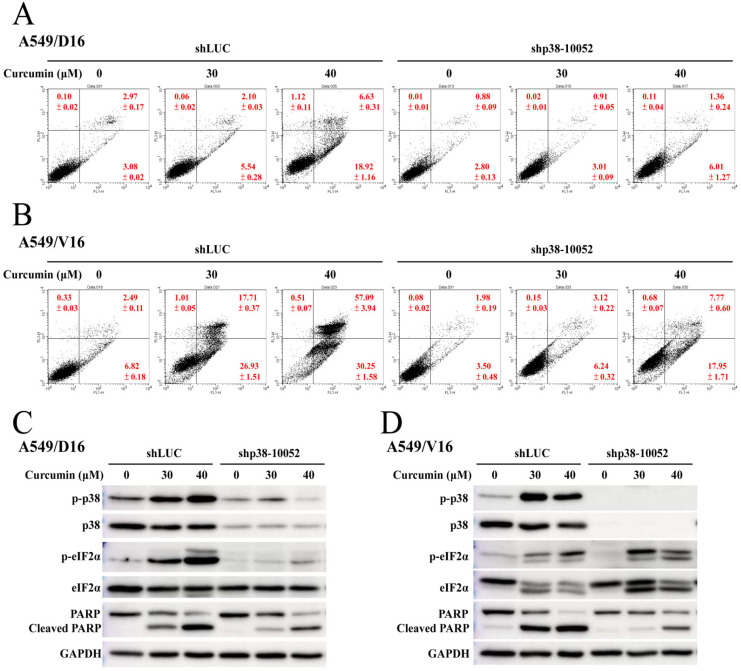
Significance of p38 MAPK knockdown on curcumin-induced apoptosis. Cells were transduced with lentiviral p38 MAPK (sh-p38-0052) vectors and control (shLUC) for 24 h. Transduced cells were selected with 2 g/mL puromycin for 48 h followed by curcumin treatment, as described in Methods. (**A**) A549/D16 and (**B**) A549/V16 cells were analyzed by flow cytometry for curcumin-induced apoptosis. Protein expression levels of p38 MAPK, phosphorylated-p38 MAPK (p-p38), eIF2-α, p-eIF2-α and PARP were detected with western blots from the cells of (**C**) A549/D16 and (**D**) A549/V16.

**Figure 8 ijms-23-08248-f008:**
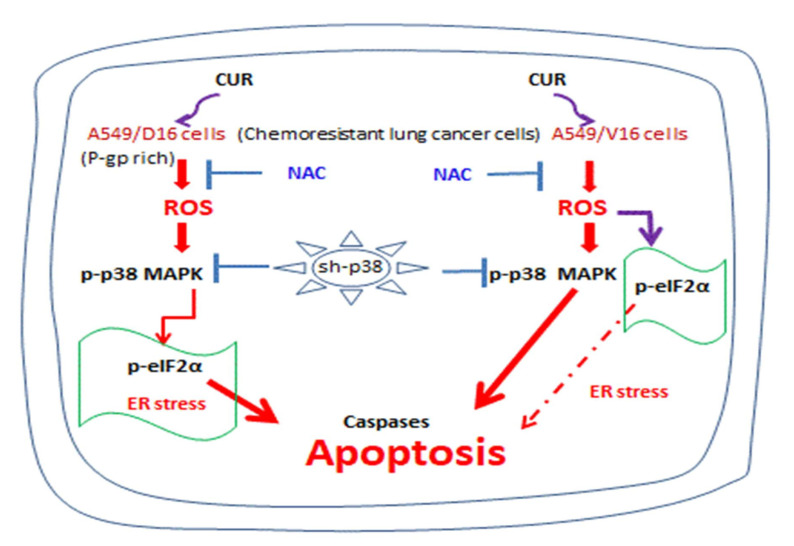
Schematic diagram of curcumin-induced ROS and apoptosis in chemoresistant lung cancer A549 cells. Curcumin (CUR) induced ROS was prevented when chemoresistant A549/D16 and A549/V16 cells were pretreated with NAC. Augmented ROS activates p38 MAPK, followed by increased ER stress demonstrated by eIF2-α phosphorylation in A549/D16 cells. Therefore, curcumin treatment resulted in caspase-dependent apoptosis. Knockdown of the p38 kinase inhibits curcumin-induced apoptosis. Under similar treatment, eIF2α phosphorylation is not related to activation of p38 MAPK by ROS in A549/V16 cells.

## Data Availability

The data used and/or analyzed during the current study are available from the corresponding author on reasonable request.

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
