# Peer review of "Curcumin Induces Apoptosis of Chemoresistant Lung Cancer Cells via ROS-Regulated p38 MAPK Phosphorylation"

_ijms, 2022, doi:10.3390/ijms23158248_

Round 1
Reviewer 1 Report
The article "Curcumin induces apoptosis of chemoresistant lung cancer cells via ROS-regulated p38 MAPK phosphorylation" explains through studies associated with chemoresistant lung cancer cells and how Curcumin induces apoptosis through ROS-regulated p38 MAPK phosphorylation.
The work represents extensive studies associated with pathway activation (JAK, ERK, p38, elf2alpha, PARP) and there phosphorylation studies with AD549/D16 and AD549/V16 and apoptosis induced through MTT assays, propidium iodide PI/Annexin V staining by flow cytometry and its detection by western blotting. Chemoresistant cancer cells were also treated with Curcumin to study apoptosis with and without pan-caspase inhibitor, ERK inhibitors, p38 kinase inhibitors, p38 MAPK knockdown.
This proves the association of Curcumin in inducing apoptosis in cehmoresistant lung cancer cells. But with respect to the direct effect of curcumin in consideration, the previous reports explains lots of other factors by which curcumin can influnce other pathways in the abscence of the targets the authors have reported.
One article by Gupta et al., 2011, elearly explains a lot about multitargeting of curcumin for various applications (Reference below)
Gupta, S. C., Prasad, S., Kim, J. H., Patchva, S., Webb, L. J., Priyadarsini, I. K., & Aggarwal, B. B. (2011). Multitargeting by curcumin as revealed by molecular interaction studies. Natural product reports, 28(12), 1937–1955. https://doi.org/10.1039/c1np00051a
Considering the broad range of activity associated with Curcumin, it is essential for the authors to explain how other pathways (in absence of those considered in this study) may affect the chemoresistant lung cancers under consideration. Also, receptor mediated inhibition of SARS-CoV2 S protein and ACE2 receptor (Jena et al., 2021)
Jena, A. B., Kanungo, N., Nayak, V., Chainy, G., & Dandapat, J. (2021). Catechin and curcumin interact with S protein of SARS-CoV2 and ACE2 of human cell membrane: insights from computational studies. Scientific reports, 11(1), 2043. https://doi.org/10.1038/s41598-021-81462-7 Considering this development, it is important to combine the interactions previously reported with those reported in this study for properly understanding the pathways the crucumin may affect. The work consideraibly validates the reported findings but to get a clearer picture of the direct effect of curcumin on chemiresistant lung cancer cells, it is essential to compare this findings with previously reported works. Apart from that the article is well organised, reporting important findings associated with the natural product derived compounds for targeting cancer and microorganisms and is suitable for publication following changes previously mentioned
Author Response
Reviewer 1
As indicated in the responses that follow, we have taken the comment into account in the revised version of our manuscript.
We would like to thank you for the valuable and constructive suggestions, which have greatly improved our manuscript.
Reply: Thank you for your comments. We have added the following paragraphs into the Discussion section to answer your suggestions.
How other pathways in addition to the reported MAPK pathway may affect the chemoresistance that attenuated by CUR? We believe that CUR diminishes cancer stemness also play a critical role on chemoresistance. The potential of CUR to regulate the growth of cancer stem cell (CSC) has been reviewed by Sordillo and Helson (49). Suppression of the cytokines, interleukin (IL)-6, IL-8, IL-1, and CSC pathways, such as Wnt, Notch, Hedgehog and FAK by CUR were indicated. It also has been reported that CUR inhibited lung cancer stem cell traits and prohibited tumorsphere formation (50). In a recent report, it has been shown CUR (125 nM) treatment reduced the sphere formation ability at the concentrations would not affect the cell viability of A549 cells and normal pulmonary epithelial cells (51). Therefore, the multi-potential of CUR with anticancer abilities warrants further experimental study to identify its optimal application.
How CUR interacts with each specific signal pathway? Moustapha et.al. (52) have analyzed that Huh-7 liver cancer cells treated with CUR (up to 80 μM) for 5 min and followed by intracellular CUR concentration measurement. Their data have shown that CUR enters cells rapidly to a final ratio between external added CUR and the intracellular concentration of 1/20. It is corresponding to 1μM of CUR as intracellular concentration for 20μM added externally. Later, same group of researchers reported that CUR localization at the endoplasmic reticulum but not on the mitochondrial network cause an unfolded protein response (UPR) and affect calcium status (53). Other ways of molecular interaction of CUR with various cellular proteins has been nicely reviewed by Gupta et. al. (54). Interestingly, a computational study for the possibility of CUR to bind the coronavirus (SARS-CoV2) viral spike protein (S Protein) and the cognate host cell receptor angiotensin-converting enzyme 2 (ACE2) has been reported (55). By using molecular simulation study, CUR directly binds to the receptor binding domain (RBD) of viral S Protein and host ACE2 thus interferes the formation of S Protein-ACE2 complex. The data provide a potential link of CUR interacts with membrane receptor for further signaling.
Sincerely,
Gwo-Tarng Sheu, Ph.D
Professor
Chung Shan Medical University,
Institute of Medicine
Reviewer 2 Report
Title: "Curcumin induces apoptosis of chemoresistant lung cancer cells via ROS-regulated p38 MAPK phosphorylation” Authors: Ming-Fang Wu, Yen-Hsiang Huang, Ling-Yen Chiu, Shur-Hueih Cherng, Gwo-Tarng Sheu, Tsung-Ying YangComments:
In this study, the authors investigated whether curcumin had a cytotoxic effect on A549 lung cancer cells treated with chemotherapeutic agents that had different P-gp expression. The authors showed that curcumin significantly induced ROS and led to apoptosis of chemoresistant lung cancer cells, regardless of P-gp expression. Knockdown of p38 MAPK, but not by its inhibitor, protected these chemoresistant cells from curcumin-induced apoptosis.
Major points:
1: Title/Abstract/Introduction
2: In my opinion, the title cannot be left as it is, since the results obtained in 2.5. and 2.6. are not meaningful enough. This is also supported by the vague statements in the discussion (page 12, lines 306-314 and page 13, lines 336-338).
3: Page 1, lines 27-37: Here the summary is written in a confusing way and should be worded in a clearer and more structured way.
4: Page 2, lines 76-92: Here the introduction is confusingly worded. The statements that are relevant to this paper should be explained in an understandable way.
5: Page 2, lines 94-99: The results of the paper should not be prefaced in the introduction.
Results/Discussion/Conclusion.
6: Page 3, lines 112-114: Section 2.1. mentions that parental A549 cells were used as controls in all experiments. The comparison is only shown in Figure 1A and 1B and should definitely be shown as a comparison for each result.
7: Page 4, Figure 1A+B: The legend needs to be adjusted because it is not possible to distinguish between the dashed and non-dashed lines. Also, the long-term effect of chemotherapeutic agents on cell types without curcumin treatment is missing as a comparison.
8: Page 5, Figure 2C+D: A blot with cleaved caspase 7 should also be shown here to validate the inhibitor and as a comparison with Figure 1.
9: Page 6, line 183-184: The conclusion of each result should refer to a new finding.
10: Figure 4-7: There are differences in the Western blot bands between the two cell types studied. These should be mentioned in the text of the results.
11: The curcumin dose chosen seems very high to me. For example, wouldn't tumor cells die if treated with 40µM curcumin for 48 hours?
12: I miss overall a tangible, pictorial underpinning of the results, for example LM, IF or EM.
13: Page 15, Conclusions: It does not fit the discussion (line 274) to define curcumin here as a pro-oxidant (line 403) as it has a regulatory effect depending on the cell situation. Overall, the results presented are too weak to draw the conclusions stated in the conclusion.
Author Response
Reviewer 2
As indicated in the responses that follow, we have taken the comment into account in the revised version of our manuscript.
We would like to thank you for the valuable and constructive suggestions, which have greatly improved our manuscript.
Point 1&2:
Reply: When the MAPK singling pathway (ERK, p38 and JNK) were investigated on CUR treated sublines, the levels of phosphorylated ERK and p38 MAPK, but not JNK, were significantly enhanced and NAC pre-treatment prevent ERK and p38 hyperphosphorylation. Therefore, we tried to block the ERK and p38 hyperphosphorylation with their inhibitors and found anti-apoptotic effect was not obtained by pretreating either with ERK nor p38 MAPK inhibitors. In Fig 5, ERK inhibitor (U0126) effectively blocked ERK hyperphosphorylation but failed to reduce CUR-induced apoptosis. Therefore, ERK activation was not associated with CUR-induced apoptosis. When p38 MAPK inhibitor (SB203580) was tested with similar conditions, SB203580 failed to reduce CUR-induced apoptosis either (Fig 6). When we carefully compared the inhibition efficiency of SB203580 on p38 phosphorylation, it can be seen that phosphorylation form of p38 (p-p38) was not reduced nor enhanced when cells were treated with CUR (Fig 6 C and D). We thought it may require a better method to block p38 phosphorylation in order to detect its phosphorylation effect on apoptosis. Therefore, p38-specific shRNA was applied to repeat the experiments as shown in Fig 7. The data clearly demonstrated that when p38 MAPK was absent, CUR-induced apoptosis was significantly prevented.
Point 3:
Reply: We have modified the abstract to make a better summary.
Point 4:
Reply: Sorry to give you trouble to read through this section, we have modified the paragraph to make it clear for introduction.
Point 5:
Reply: Thank you, we have deleted the sentence and modified the paragraph accordingly.
Point 6:
Reply: The reason of A549 cells used in drug sensitivity assay (Fig 1A and B) was to demonstrate that only parental A549 cells but not A549/D16 and A549/V16 sublines response to DOC and VCR. Actually, the effect of CUR on A549 cells have been demonstrated and reported by Yao et.al. (Ref 27: Curcumin induces the apoptosis of A549 cells via oxidative stress and MAPK signaling pathways. Int J Mol Med 36: 1118-1126, 2015) and Liu et.al. (Ref 35: Antitumor activity of curcumin by modulation of apoptosis and autophagy in human lung cancer A549 cells through inhibiting PI3K/Akt/mTOR pathway. Oncol Rep 39: 1523-1531, 2018). To avoid copyright issue, we focused at chemoresistant A549 cells in this study.
Point 7:
Reply: We have redone the Figure 1 to show the dashed line clearly to prevent confusing. Since the A549/D16 and A549/V16 cells were stably resistant to 16 nM of DOC and VCR, the growth of sublines have not affect in the present of DOC/VCR.
Point 8:
Reply: Cleavage of PARP by caspases is considered to be a hallmark of apoptosis. Almost all caspases are known to modify PARP-1 in vitro. In vivo, both caspase 3 and caspase 7 cleaves PARP-1. Therefore, we showed the pattern of original (116 kDa) and cleaved (89 kDa) PARP proteins to support the pan-caspase activity. Practically, if apoptosis was not significantly observed by PI/Annexin V staining of flow cytometry analysis, the large fragment of cleaved caspase 7 (20 kDa) protein is not easy to be detected. Otherwise, the results in Fig 2 should have enough information to demonstrate that CUR induces caspase dependent apoptosis.
Point 9:
Reply: Yes, thank you for the comment. Our data show NAC blocks the apoptosis in both chemoresistant sublines when cells are exposed to CUR. The data indicates that antioxidants application on cancer patients must carefully monitored when patient is under CUR administration.
Point 10:
Reply: You mean the data of (Figure 7C, D). We have written in page 11 of result section 2.6 as following: The data showed the protein levels of p38 MAPK were merely detected in both sublines and the phosphorylation of eIF2α was blocked in the A549/D16 cells but not in the A549/V16 cells. When p38 MAPK knockdown was achieved, the cleaved PARP levels were reduced that coordinated with the CUR-induced apoptosis. Further, p38 MAPK was found to be an upstream mediator of eIF2α phosphorylation in A549/D16 cells but this activity was not found in A549/V16 cells.
Point 11:
Reply: The answer is: Yes, under 40 µM of CUR exposure, A549 and chemoresistant sublines would be all die of apoptosis. We pointed out that (1) CUR increases the DOC/VCR toxicity of A549 cells in a dose-dependent manner (Figure 1), which is a chemotherapy enhancement effect. (2) CUR alone can reduce DOC- and VCR-resistant cell survival; the combination of chemotherapy with CUR has little effect in further decreasing the survival of chemoresistant cells. Therefore, no need of chemotherapy if chemoresistance has been established; CUR alone is a better choice to control drug resistant cancer cells to avoid the side effect generated from chemotherapy. Other reported data (Ref 27 and Ref 35) using lung cancer cells also used equivalent dose of CUR for their experiments.
Point 12:
Reply: Although the LM, IF or EM were not found in our text, thank you for this correction, we already added into the figure legend of Fig 1. (lower right square, LR; upper right square, UR; upper left square, UL).
Point 13:
Reply: According to the data showed in this study, CUR concentration higher than micromole has ROS induction activity that fit to the pro-oxidant classification. To follow your comment, we modified the sentence as: The general summary provided in Fig 8 concludes CUR, at a high concentration, is an effective pro-oxidant molecule to generate high levels of intracellular ROS to induce caspase-dependent apoptosis of chemoresistant lung cancer cells.
Sincerely,
Gwo-Tarng Sheu, Ph.D
Professor
Chung Shan Medical University,
Institute of Medicine
Round 2
Reviewer 2 Report
The authors have satisfactorily addressed the concerns raised in the original version. The revised version is significantly improved. No further concerns.